# Leveraging Asynchronicity in Gradient Descent for Scalable Deep Learning

**Jeff Daily, Abhinav Vishnu, Charles Siegel**
Pacific Northwest National Laboratory
902 Battelle Blvd
Richland, WA 99352
{jeff.daily,abhinav.vishnu,charles.siegel}@pnnl.gov

## Abstract

In this paper, we present multiple approaches for improving the performance of gradient descent when utilizing mutiple compute resources. The proposed approaches span a solution space ranging from equivalence to running on a single compute device to delaying gradient updates a fixed number of times. We present a new approach, asynchronous layer-wise gradient descent that maximizes overlap of layer-wise backpropagation (computation) with gradient synchronization (communication). This approach provides maximal theoretical equivalence to the *de facto* gradient descent algorithm, requires limited asynchronicity across multiple iterations of gradient descent, theoretically improves overall speedup, while minimizing the additional space requirements for asynchronicity. We implement all of our proposed approaches using Caffe – a high performance Deep Learning library – and evaluate it on both an Intel Sandy Bridge cluster connected with Infini-Band as well as an NVIDIA DGX-1 connected with NVLink. The evaluations are performed on a set of well known workloads including AlexNet and GoogleNet on the ImageNet dataset. Our evaluation of these neural network topologies indicates asynchronous gradient descent has a speedup of up to 1.7x compared to synchronous.

## 1 Introduction

Deep Learning (DL) algorithms are a class of Machine Learning and Data Mining (MLDM) algorithms, which use an inter-connection of *neurons* and *synapses* to emulate the computational structure of a mammalian brain. DL algorithms have demonstrated resounding success in many computer vision tasks and science domains such as high energy physics, computational chemistry and high performance computing use-cases. Several DL implementations such as TensorFlow, Caffe, Theano, and Torch have become available. These implementations are primarily geared towards compute nodes that may contain multi-core architecture (such as Intel Xeon/KNC/KNL) and or many-core architectures (GPUs).

DL algorithms are under-going a tremendous revolution of their own. Widely used DL algorithms such as Convolutional Neural Networks (CNNs) and Recurrent Neural Networks (RNNs) are computationally expensive. Their computational requirements are further worsened by: 1) Very deep neural networks such as recently proposed 1000-layer complex Residual Networks (*ResNet*), 2) Increasing volume of data produced by simulations, experiments and handheld devices. An important solution to these problems is the design and implementation of DL algorithms that are capable of execution on distributed memory large scale cluster/cloud computing systems. A few distributed DL implementations such as CaffeonSpark, Distributed TensorFlow, CNTK, Machine Learning Toolkit on Extreme Scale (MaTEx), and FireCaffe have become available. Implementations such as CNTK, FireCaffe and MaTEx use MPI (Gropp et al., 1996; Geist et al., 1996) – which makes them a natural fit for high-end systems.

DL algorithms primarily use *gradient descent* – an iterative technique in which the weights of *synapes* are updated using the difference between the *ground truth* (actual value) and the *predicted value* (using the current state of the neural network). The larger the difference, the steeper the de-

scent to a minima (a low value of minima generates the solution). An important type of gradient descent is *batch gradient descent* – where a random subset of samples are used for iterative *feed-forward* (calculation of predicted value) and *back-propagation* (update of synaptic weights). A small batch is prone to severe pertubations to the descent, while a large batch results in slow convergence. Hence, a data scientist tends to use a fairly average batch – which finds the balance between these two conflicting metrics.

A large scale parallelization of gradient descent must maximize the equivalence to the default algorithm, such that the convergence property is maintained. Consider a scenario where a batch ($b$) in the original algorithm is split across multiple compute nodes ($n$) – an example of *data parallelism*. To provide equivalence to the default algorithm, the batch must be split equally to $\frac{b}{n}$, although the communication which would require an all-to-all reduction would increase as $\Theta(\log n)$. Naturally, as $n$ is increased and $b$ is held constant (*strong scaling*), this becomes prohibitive, whereas keeping the batch size per node $b/n$ constant (*weak scaling*) increases the convergence time.

Several researchers have proposed methods to alleviate the communication requirements of distributed gradient descent. Parameter-server based approaches use a server to hold the latest version of the model while clients send computed gradients and request the latest model. This approach has been proposed and extended by several researchers. While theoretically this provides $O(1)$ time-complexity since all batch updates can be computed simultaneously, this approach fails to scale beyond a few compute nodes when considering the time to convergence relative to having run the computation on a single device. Others have proven divergence from the original algorithm. Remote Direct Memory Access (RDMA) based approaches have been proposed, but they also diverge from the original algorithm. Several other implementations are primarily geared towards shared memory systems, and address the thread contention issue for gradient descent.

Our objective is to design a non-parameter-server based technique, which maximizes the equivalence to the default algorithm, while leveraging high performance architectures – including computational units such as GPUs and high performance interconnects such as InfiniBand, Intel Omni-path architectures by using MPI.

## 1.1 CONTRIBUTIONS

Specifically, we make the following contributions in this paper:

- We design a baseline asynchronous gradient descent, which delays the gradient updates of the entire model by one or more iterations adaptively on the basis of available overlap and user-defined input.

- We propose a layer-wise gradient descent method, which overlaps weight updates of a layer with inter-node synchronization of other layers. The proposed method is exactly equiavalent to the default sequential algorithm.

- We implement our approaches and other baseline techniques using the Machine Learning Toolkit for Extreme Scale (MaTEx), which consists of a distributed memory implementation of Caffe using MPI (Gropp et al., 1996; Geist et al., 1996).

- We evaluate our approaches and other baseline implementations on a large scale CPU-based InfiniBand cluster as well as on NVIDIA's DGX-1 multi-GPU system. We use several well studied datasets and DNN topologies such as ImageNet (1.3M images, 250GB dataset) with AlexNet and GoogleNet DNNs.

Our evaluation indicates the efficacy of the proposed approach. Specifically, the best asynchronous approach is up to 1.7x faster than the synchronous approach while achieving up to 82% parallel efficiency.

The rest of the paper is organized as follows: In section 2, we present related work of our proposed research. We present the background in section 3, followed by an in-depth solution space in section 4. In section 6, we present a detailed performance evaluation of asynchronous gradient descent, and conclusions with future directions in section 7.

## 2 RELATED WORK

Batch gradient descent is the most widely used algorithm for training Deep Learning models. This algorithm has been implemented several times for sequential, multi-core and many-core systems such as GPUs. The most widely used implementations are Caffe (Jia et al., 2014) (CPUs/GPUs), Warp-CTC (GPUs), Theano (Bastien et al., 2012; Bergstra et al., 2010) (CPUs/GPUs), Torch (Collobert et al., 2002) (CPUs/GPUs), CNTK (Agarwal et al., 2014) (GPUs and Distributed Memory using MPI) and Google TensorFlow (Abadi et al., 2015) which use nVIDIA CUDA Deep Neural Network (cuDNN).

Caffe is one of the leading software tools for training and deploying deep learning algorithms, and it can be used to develop novel extensions to these algorithms such as the ones described below. Caffe supports execution on a single node (connected with several GPUs) and a version has been implemented that takes full advantage of Intel systems. While the research described below was performed using Caffe, the extensions can be applied to Tensorflow as well.

Caffe (and other deep learning software) is also equipped with several optimizations designed to avoid significant problems in training deep networks. The *vanishing gradient* problem (Bianchini & Scarselli, 2014) causes deep networks to fail to learn much at all in the early layers, and was solved in (Hinton & Osindero, 2006) and (Bengio et al., 2007) where it was shown that a network could be trained one layer at a time with *autoencoders* (Hinton & Salakhutdinov, 2006), and then put together to form a single network (Vincent et al., 2010). Another optimization that helps to solve this problem is switching from *sigmoidal neurons* to *rectified linear neurons*.

The problem of accelerating gradient descent, especially disctributed across compute resources, is of interest to many researchers. Approaches generally fall into two categories, whether or not they are equivalent to having run using a single compute device; utilizing a single compute device necessarily computes gradient updates and applies them immediately to the model. Further, the gradient updates can be classified as either synchronous or asynchronous depending on whether the communication of the gradients can be overlapped with any computation of the gradients. For example, the DistBelief parameter server approach (Dean et al., 2012) computes gradient updates asynchronously based on an out-of-date copy of the model and applies them to the latest model. Though this is not equivalent to having run on a single device, it is able to process samples much faster.

Chen et al. (2016) revisit asynchronous gradient descent and propose a few synchronous variants in order to impove time to convergence. Notably, they show that waiting for all workers to complete, aggregating the gradients, and applying the gradients to the same common model (thereby each worker has a copy of the latest model) provides a good time to convergence while also leveraging multiple compute devices. Their approach is where this paper begins while additionally proposing approaches ranging from synchronous to parameter server variants.

## 3 FUNDAMENTALS

### 3.1 NEURAL NETWORKS

Machine Learning algorithms designed to emulate the computational structure of the brain to model data are called "Neural Networks." The basic unit of a neural network is the *neuron* and neurons are connected to one another via *synapses*.

#### 3.1.1 BACKPROPAGATION

Neural networks are trained through an algorithm called *backpropagation*. This is a means of computing gradients layer by layer to implement the *gradient descent algorithm*'s update rule of

$$\mathbf{w}' = \mathbf{w} + \lambda \nabla_{\mathbf{w}} C \tag{1}$$
$$\mathbf{b}' = \mathbf{b} + \lambda \nabla_{\mathbf{b}} C \tag{2}$$

where $\mathbf{w}$ are the weights, $\mathbf{b}$ the biases, $\lambda$ the learning rate, and $C$ is a cost function to be optimized, usually square error or cross-entropy. This rule is often replaced by a slightly more complex rule, such as Adaptive Gradient Descent (AdaGrad) (Duchi et al., 2011) or Momentum (Qian, 1999).

To compute the gradients, we set $W^{(\ell)}$, $b^{(\ell)}$ the weights and biases for each layer, $z^{(\ell+1)} = W^{(\ell)}a^{(\ell)} + b^{(\ell)}$ and $a^{(\ell)} = \sigma(z^{(\ell)})$, where $\sigma$ is the activation function. Let $n_\ell$ represent the number of layers. Then, we use Algorithm 1.

---

**Algorithm 1** Back Propagation

---

1: **input:** Data $X \in \mathbb{R}^{n \times p}$ and labels $Y \in \mathbb{R}^{n \times \ell}$
2: **for** $i$ from 1 to $n$ **do**
3: Compute all $z^{(\ell)}$ and $a^{(\ell)}$.
4: $\delta^{(n_\ell)} = -(y - a^{n_\ell}) \odot \sigma(z^{(n_\ell)})$
5: **for** $\ell$ from $n_\ell - 1$ to 2 **do**
6: $\delta^{(\ell)} = W^\ell \delta^{(\ell+1)} \odot \sigma'(z^{(\ell)})$
7: **end for**
8: $\nabla_{W^{(\ell)}} C = \delta^{(\ell+1)} a^{(\ell)T}$
9: $\nabla_{b^{(\ell)}} C = \delta^{(\ell+1)}$
10: **end for**

---

Although there are several nonlinear activation functions in common use, the networks examined in this paper only include rectified linear units (ReLU) where $\mathrm{ReLU}(x) = \max(0, x)$.

### 3.2 CAFFE

Caffe (Jia et al., 2014) is one of the leading software packages for building and training neural networks. It provides abstractions for a wide range of topologies and for training them with many different types of optimizers. Caffe provides abstractions for operations on multi-dimensional arrays (tensors) which are essential for implementing Deep Learning algorithms. From an input tensor, an output tensor, and tensors for each hidden layer, Caffe constructs a computational graph that manages these tensors and their updates as a single object. Caffe is particularly useful for researchers, because it is heavily optimized and can be modified through an open source C++ backend.

As Caffe's runtime is implemented in C++, it can extract native performance from the computation environment it is run on. Furthermore, Caffe abstracts GPU computations, leveraging nVIDIA CUDA Deep Neural Network Library (cuDNN) for the task. We have modified this code for distributed memory computation on large scale systems using MPI to natively use network hardware for optimal performance. The base, synchronous implementation is similar to FireCaffe (Iandola et al., 2015), another distributed memory implementation of Caffe. Further modifications are described in Section 4.

There are three phases of computation within Caffe that pass over the enumerated layers of the network. First, the forward pass computes the output result given the samples from the input batch, starting at the first layer. Next, starting at the last (output) layer, based on the difference between the output result and the ground truth, the backward pass uses the backpropagation technique to compute the gradients for each layer. Lastly, one final pass is made over the network to apply the gradients to the weights and biases before starting the process over again with the next batch.

## 4 SOLUTION SPACE

The goal of improving gradient descent is to accelerate the time to solution without sacrificing the accuracy of the model. The base case to consider is then computing and applying gradients one batch at a time on a single compute device. One way to accelerate the computation while also maintaining equivalence to the sequential is to use data parallelism. Data parallelism is where the traditional batch is further subdivided into equally-sized mini-batches, each mini-batch is computed on separate devices, then the gradients resulting from each mini-batch is averaged together. Since each gradient update is itself an average, taking the average of the mini-gradients results in an update that is effectively the same as having computed the original batch size. This is called the *effective batch size*. Data parallelism is the approach we explore in this paper, attempting many ways of hiding the latency of the gradient communication that occurs between compute devices. We use MPI to communicate the gradients.

Caffe provides callback methods in its C++ interface that interject user-defined functionality into key phases of the computation (see 3.2). Specifically, one user-defined function is executed immediately before the foward pass when the batch computation begins. The other user-defined function executes after the backward pass finishes, but before the application of the gradients to the weights and biases. Additional callback functions were added to support finer-grained control over the three phases of computation. One of the additional callbacks executes after each gradient is computed during the backward phase, once per set of learnable parameters, such as the weights or biases of a given layer. Another callback function that was added is called once per learnable parameter during the apply phase, just before the gradients are applied. Lastly, a callback function was added that turns the gradient application into a task queue, requesting additional tasks in an unspecified order until all gradients have been applied.

A critical implementation detail for any of our proposed approaches is to make sure the individual network models maintained by each compute device start from the same random initial conditions for the weights and biases. Before the first batch is computed, the weights and biases from the master process are copied (broadcast) to the other processes. That way any gradients that are computed, when averaged together, are based on the same initial conditions.

## 4.1 SYNCHRONOUS GRADIENT DESCENT

Similar to what Chen et al. (2016) proposes and what is implemented in FireCaffe (Iandola et al., 2015), synchronous gradient descent averages the gradients from each mini-batch together before applying them, forming one complete batch at a time. The way this is implemented in Caffe is to use the callback function that executes when all gradients are ready to be applied. During this callback, MPI_Allreduce is used to sum the gradients, placing the same resulting sum on each compute device. This function is blocking, meaning it returns control back to Caffe only after the sum is computed across all devices. Since the result is a sum and not the intended average, it is then scaled down based on the number of compute devices in use. It is important to note that the reduction operation can be performed in-place, meaning it can use the memory location directly holding the gradient without performing any costly memory copies, especially for networks with a large number of parameters such as AlexNet. This approach also has the important quality that the gradients are averaged *after* they have been used by each layer of the backpropagation, preserving the importance of any activations within the network against the mini-batch instead of against the effective batch.

## 4.2 LAYER-WISE GRADIENT DESCENT

Chen et al. (2016) proposes the pipelining of gradient computation and application. For example, the gradients of upper layers can be concurrently applied while computing the gradients of lower layers. This approach must be done carefully to maintain equivalence with the sequential base case. We make the observation that gradients can be averaged as soon as they are computed during the backward phase, instead of waiting for all gradients to be computed. However, adjacent layers will use and/or update the gradients of layers that have otherwise finished computing their gradients. This implies the averaging of the gradients must be performed on a copy of the gradients rather than in-place. Further, the averaging of the copied gradients must finish before they can be applied.

We utilize a background thread of computation in order to perform the gradient averaging concurrent with the remaining gradient computation. This provides maximal overlap of the communication latency with useful computation. There are a few options when to apply the averaged gradients. Waiting for all communication to finish before applying all gradients is straightfoward and similar to the synchronous approach described previously, though perhaps at least some of the communication latency would be overlapped. Another approach is to wait, one layer at a time, for the gradients for a particular layer to finish averaging and then apply the gradients. It is intuitive to perform the waiting in the same order in which backpropagation was performed, from the last layer to the first layer. Lastly, since all gradient updates are independent, we can perform them in an arbitrary order. This takes advantage of the observation that not all layers have the same number of parameters, and further, the gradients for the weights and the gradients for the biases can be averaged separately; the size of the weight gradients are typically larger than the bias gradients, implying that the bias gradients will complete their communication more quickly. Since the communcation of the various parameters can finish somewhat arbitrarily based on when the communication was initiated and the

size of the communication, we can apply the gradients as soon as they complete their averaging. We evaluate these strategies in 6.

### 4.3 ASYNCHRONOUS GRADIENT DESCENT

As stated in (Chen et al., 2016), parameter server implementations suffer from poor convergence since gradient updates are calculated based on out-of-date networks. Continuing with our data parallel approach, there is a lower limit to the size of the mini-batches and therefore the number of compute devices that can be utilized. As the amount of work per compute device decreases proportional to the decreasing size of the mini-batches, there is less computation available to mask the latency of the gradient averaging across the devices. Initiating the averaging layer-wise as described above may not be enough to mitigate this problem.

We propose delaying the application of the gradients by a fixed number of iterations much smaller than the number of compute devices as would have been done in a parameter server approach. The gradients are delayed by using a concurrent communication thread and applying the gradient one, two, or three iterations later thus giving the averaging enough time to complete as needed. If the gradient needs to be delayed by one iteration, this requires one communication thread and one additional buffer to hold the gradient; delaying by two iterations requires two communication threads and two additional buffers and so on. This approach is somewhere between a parameter server (Dean et al., 2012) and the various approaches that maintain equivalency with a sequential computation.

## 5 IMPLEMENTATION DETAILS

The implementations evaluated in this paper focus on data parallelism and the averaging of gradients across compute devices. This is achieved using MPI and parallel I/O.

### 5.1 HANDLING I/O

The data parallelism is achieved by distributing datasets across compute devices, partitioning them based on the number of devices utilized; each device receives a disjoint subset of the dataset and no samples are shuffled or exchanged between compute devices outside of the gradient averaging. Caffe frequently uses a database in LMDB format for its datasets, however this format cannot be used on remote (network) filesystems or even between processes on the same host. Caffe mitigates this issue when using more than one GPU on the same host by using a single I/O reading thread and a round-robin deal of the samples to device-specific queues. Our implementations mitigate this issue by first converting an LMDB database into a netCDF file (Rew & Davis, 1990). netCDF files can be read and partitioned using parallel MPI-IO via the parallel netCDF library (Li et al., 2003).

### 5.2 DISTRIBUTED MEMORY IMPLEMENTATION USING MPI

For single-node GPU computation, using one or more GPU devices in a single host, Caffe provides a means of allocating one contiguous buffer to hold the data for the weights and biases and a second buffer to hold the gradients for each. We extended this approach for CPU hosts. A single contiguous buffer allows the non-layer-wise, i.e., network-wise gradient averages to be performed using a single MPI reduction operation. The layer-wise implementations require one MPI reduction operation per network parameter. There is a fixed cost to start a communication primitive regardless of how much data is communicated. It is sometimes beneficial to aggregate otherwise many small communication requests into a larger one.

Although Caffe provides a way of utilizing all GPUs within the host, it does not currently leverage NVIDIA's NCCL package (NVIDIA Corporation, 2015) for optimized, high-bandwidth collective communication routines. We used the NCCL equivalent to the MPI all reduction to sum gradients across GPU devices on the DGX-1 platform.

## 6 Experimental Evaluation

In this section, we present an experimental evaluation and analysis of the heuristics described in section 4.

### 6.1 Hardware Architectures

We evaluate using a CPU cluster as well as NVIDIA's speialized DGX-1 multi-GPU host system. Each node of the multi-node cluster consists of a multi-core Intel Sandybridge CPU connected via InfiniBand. We use Intel MPI 5.1.2 for performance evaluation. The heuristics are implemented in Caffe (Jia et al., 2014), specifically the intelcaffe branch designed to optimize performance on Intel CPUs.

The DGX-1 system contains 8 Pascal GPUs connected using the high-speed NVlink interconnect. For the DGX-1 evaluations, the latest version of Berkley's Caffe was modified to use the NCCL communicaiton primitives in addition to our algorithmic changes.

### 6.2 ImageNet and Network Architectures

We evaluate on two distinct network architectures trained on the ImageNet dataset. ImageNet refers specifically to the ILSVRC2015 (Russakovsky et al., 2015) dataset. This dataset consists of a training set of just under 1.3 million images of various sizes (as jpg files) divided among 1000 classes, along with a validation set consisting of 50000 images of the same type and classes. Additionally, for the competition, there is a testing set, but it is held separately and not available publicly. It is established as one of the benchmark dataset for machine learning with large datasets, and among the famous architectures that achieved record top 1 and top 5 accuracies on it are AlexNet (Krizhevsky et al., 2012) and GoogLeNet (Szegedy et al., 2015).

We evaluate on AlexNet and GoogLeNet because they are now well-established models with known training regimes and loss curves. They also demonstrate two different regimes for parallelization: AlexNet has approximately 60 million parameters that need to be communicated, whereas GoogLeNet has approximately 4 million. In contrast to the smaller amount of communication for GoogLeNet, it requires roughly twice the amount of time to process a each image than AlexNet does when communication is ignored.

### 6.3 Evaluation

Figure 1 compares the implemented approaches relative to a communication-less baseline "no comm". The effective batch sizes were 256 and 32 for AleNet and GoogLeNet, respectively. For example, using 8 compute devices for GoogLeNet uses a mini-batch size of $32/8 = 4$. The evaluation on DGX-1 were limited to 8 compute devices whereas the CPU cluster evaluation eventually hit the strong scaling limit for data parallelism.

These results show that delaying the gradient updates by one or more iterations is the most effective means of hiding the communication latency. The layer-wise approaches did not perform as well as expected. These trends were consistent across both hardware platforms.

The layer-wise approaches, though promising as equivalent to a sequential computation, were not able to complete their gradient averages quickly enough. Compared to the delayed gradient approach, this is perhaps intuitive. The delayed gradient approach is able to hide the communication latency across all three complete phases of the computation whereas the layer-wise approaches only have as long as it takes to complete the backpropagation phase. This is not enough time to complete the communication, especially as the mini-batch sizes decrease and therefore provide less work to mask the communication.

In addition to looking at the time per batch above, the rates of convergence of these heuristics must be evaluated. All of the heuristics completed training AlexNet to the standard top-1 accuracy of $\approx 54\%$ using the default AlexNet settings that come with Caffe. However, it is worth noting that at the beginning of training, they showed different loss curves showing that there is a tradeoff between number of batches per second and accuracy at a given batch as shown in Table 1.

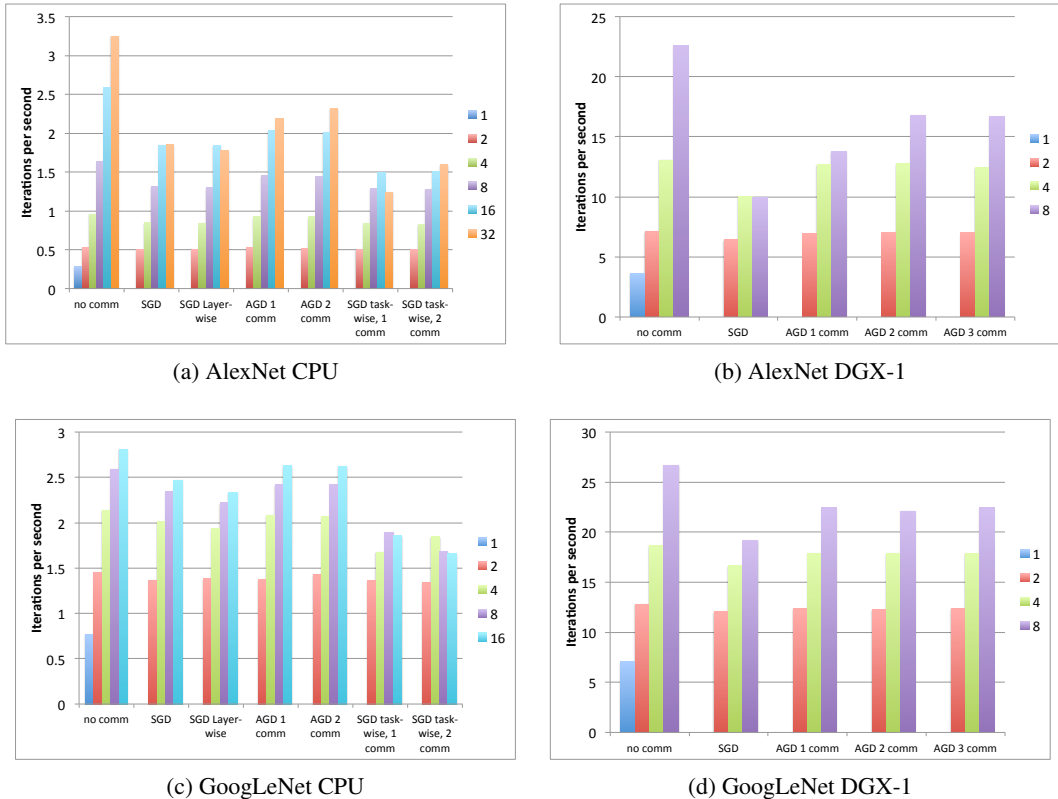

Figure 1: Evaluation of SGD and AGD approaches. Effective batch sizes were 256 and 32 for AlexNet and GoogLeNet, respectively.

Table 1: AlexNet Accuracy After Every 1000 Batches on DGX-1

| batch | 1000 | 2000 | 3000 | 4000 | 5000 |
|---|---|---|---|---|---|
| serial, 1 GPU | 0.0124 | 0.05164 | 0.10102 | 0.13432 | 0.16454 |
| SGD | 0.01116 | 0.03984 | 0.07594 | 0.10622 | 0.13052 |
| AGD, 1 comm | 0.0039 | 0.01324 | 0.02632 | 0.05076 | 0.07362 |
| AGD, 2 comm | 0.00104 | 0.00356 | 0.00636 | 0.01282 | 0.01688 |

We also evaluated whether these approaches converged in addition to just improving the number of iterations per second. All approaches evaluated managed to converge within the exepcted number of iterations. Notably, AlexNet on DGX-1 reached convergence in 11 hours using the delayed gradient approach and two communication threads using the standard AlexNet network from Caffe.

# 7 CONCLUSIONS

There is a tradeoff between maintaining equivalence to sequential methods versus leveraging the vast computational resources available for gradient descent. We find that asynchronous methods can give a 1.7x speedup while not sacrificing accuracy at the end of an otherwise identical training regime. This improvement was achieved without the need for a warm start, contrary to previously published results using parameter servers.

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
