# Peer review of "Leveraging Asynchronicity in Gradient Descent for Scalable Deep Learning"

_ICLR 2017 — rejected_

[Official Review · AnonReviewer3 · rating 3 · confidence 5 · 06 Dec 2016]
**Difficult to read paper. Lack of strong async baseline a major flaw.**

This paper is relatively difficult to parse. Much of the exposition of the proposed algorithm could be better presented using pseudo-code describing the compute flow, or a diagram describing exactly how the updates take place. As it stands, I'm not sure I understand everything. I would also have liked to see exactly described what the various labels in Fig 1 correspond to ("SGD task-wise, 1 comm"? Did you mean layer-wise?).
There are a couple of major issues with the evaluation: first, no comparison is reported against baseline async methods such as using a parameter server. Second, using AlexNet as a benchmark is not informative at all. AlexNet looks very different from any SOTA image recognition model, and in particular it has many fewer layers, which is especially relevant to the discussion in 6.3. It also uses lots of fully-connected layers which affect the compute/communication ratios in ways that are not relevant to most interesting architectures today.

[Public Comment · Xinghao Pan · 12 Dec 2016]
**Chen et al., and wallclock time evaluations**

Full disclosure: I am an author on the follow-up paper to Chen et al. (2016).

Although the paper cites Chen et al. (2016) as a starting point of the proposed approach, it does not mention the key message of Chen et al. (2016) of using backup workers to hide straggler effects in synchronous gradient descent. Were backup workers used as part of the synchronous approaches?

Also, the paper does not evaluate the wallclock time to reach convergence or a given accuracy. Even though the AGD approach has more iterations per second, it also leads to poorer convergence (as acknowledged in Table 1). As a specific example, to reach 0.01688 accuracy requires <2000 iterations of SGD, <3000 iterations of AGD (1 comm), and 5000 iterations of AGD (2 comm). Hence, for AGD to be faster than SGD in wallclock time, AGD must have at 1.5x iterations per second relative to SGD, which does not appear to the case from Figure 1. Of course, this example is only for a low accuracy of 0.01688, and the proposed AGD could very well reach the final convergence of 54% in lesser time. It would be very helpful if the authors could provide that information.

[Official Review · AnonReviewer1 · rating 3 · confidence 4 · 14 Dec 2016]
**Lacks Strong Baselines and Wall-Time Results**

The authors present methods to speed-up gradient descent by leveraging asynchronicity in a layer-wise manner.

While they obtain up-to 1.7x speedup compared to synchronous training, their baseline is weak. More importantly, they dismiss parameter-server based methods, which are becoming standard, and so effectively just do not compare to the current state-of-the-art. They also do not present wall-time measurements. With these flaws, the paper is not ready for ICLR acceptance.

[Official Review · AnonReviewer2 · rating 5 · confidence 4 · 16 Dec 2016 (modified: 17 Dec 2016)]
**review for Leveraging Asynchronicity in Gradient Descent for Scalable Deep Learning**

This paper describe an implementation of delayed synchronize SGD method for multi-GPU deep ne training.
Comments
1) The described manual implementation of delayed synchronization and state protection is helpful. However, such dependency been implemented by a dependency scheduler, without doing threading manually.
2) The overlap of computation and communication is a known technique implemented in existing solutions such as TensorFlow(as described in Chen et.al) and MXNet. The claimed contribution of this point is somewhat limited.
3) The convergence accuracy is only reported for the beginning iterations and only on AlexNet. It would be more helpful to include convergence curve till the end for all compared networks.

In summary, this is paper implements a variant of delayed SyncSGD approach. I find the novelty of the system somewhat limited (due to comment (2)). The experiments should have been improved to demonstrate the advantage of proposed approach.

[Final Decision · Program Chairs · 06 Feb 2017]
**ICLR committee final decision**

A summary of strengths and weaknesses brought up in the reviews:
 
 Strengths
 -Paper presents a novel way to evaluate representations on generalizability to out-of-domain data (R2)
 -Experimental results are encouraging (R2)
 -Writing is clear (R1, R2)
 
 Weaknesses
 -More careful controls are needed to ascertain generalization (R2)
 -Experimental analysis is preliminary and lack of detailed analysis (R1, R2, R3)
 -Novelty and discussion of past related work (R3)
 
 The reviewers are in consensus that the idea is exciting and at least of moderate novelty, however the paper is just too preliminary for acceptance as-is. The authors did not provide a response. This is surprising because specific feedback was given to improve the paper and it seems that the paper was just under the bar. Therefore I have decided to align with the 3 reviewers in consensus and encourage the authors to revise the paper to respond to the fairly consistent suggestions for improvement and re-submit.